# Polyols and Polyurethane Foams Obtained from Mixture of Metasilicic Acid and Cellulose

**DOI:** 10.3390/polym14194039

**Published:** 2022-09-27

**Authors:** Jacek Lubczak, Renata Lubczak, Ewelina Chmiel-Bator, Marzena Szpiłyk

**Affiliations:** Faculty of Chemistry, Rzeszów University of Technology, Al. Powstańców Warszawy 6, 35-959 Rzeszów, Poland

**Keywords:** metasilicic acid, cellulose, hydroxyalkylation, polyol, polyurethane foams

## Abstract

Hydroxyalkylation of the mixture of metasilicic acid and cellulose with glycidol and ethylene carbonate leads to a polyol suitable to obtain rigid polyurethane foams. The composition, structure, and physical properties of the polyol were studied in detail. The obtained foams have apparent density, water absorption, and polymerization shrinkage, as well as heat conduction coefficients similar to conventional, rigid polyurethane foams. The polyols and foams obtained from environmentally unobtrusive substrates are easily biodegradable. Additionally, the obtained foams have high thermal resistance and are self-extinguishing. Thermal exposure of the foams leads to an increase of the compressive strength of the material and further reduces their flammability, which renders them suitable for use as heat insulating materials.

## 1. Introduction

Silicon is an element improving the mechanical resistance of polyurethane foams (PUFs) if it is incorporated into the structure of polymeric material [1]. It has been shown that incorporation of reactive silicone compounds into the structure of PUF can also increase thermal and flame resistance of PUFs [2,3,4,5]. Firstly, the presence of silicon in a polymer reduces the percentage of other elements responsible for flaming and, secondly, during the ignition of the material, the silica of the flaming composite is gathered on the surface to form a heat barrier. The silica flame retardants are environmentally friendly because they do not emit corrosive smoke while flaming [4,5]. 

One of the silicon sources used as thermal resistance modifiers is silsesquioxanes. They have hydroxyl peripheral groups which can be used to react with isocyanate groups of urethane prepolymers to give polyurethanes of good thermal resistance [6,7,8,9]. It has been evidenced that hybrid PUFs based on polyurethane and polysiloxanes are better heat insulators than PUF itself [10].

There are not many reports on polyols with incorporated silicon. The synthesis of oligoetherol suitable for further obtaining PUF has been described in [1]. It was obtained by hydroxyalkylation of metasilicic acid with glycidol (GL) and then with ethylene carbonate (EC) at the H_2_SiO_3_:GL:EC molar ratio on polyol equal to 1:4:3 (Figure 1):

The PUFs obtained from it had similar properties to classic PUFs but showed increased thermal resistance and also considerable improved mechanical resistance. They can stand long lasting heat exposure at 175 °C. After thermal exposure they also have remarkably higher compressive strength than before thermal treatment [1].

Cellulose is a widely spread carbohydrate in nature. This biopolymer is formed from carbon dioxide and water during photosynthesis in plants [11]. Due to the ecologically justified withdrawal of fossil resources, the polysaccharides are widely tested as substrates for the chemical industry. They are used to obtain polyols which can, at least partially, replace polyols obtained from petrochemical raw materials. Cellulose (I) is glucopyranose linear polymer with meres linked by β-1,4-glycoside bonds (Figure 2). Within the cellulose meres there are intramolecular hydrogen bonds between C3-OH and ring oxygen atoms. Moreover, the packing of solid particles is driven by inter-chain hydrogen bonds, which renders the solid cellulose rigid and having very low solubility [12], and consequently very low reactivity. Finally, it is practically insoluble in water or organic solvents.

It is hygroscopic. Water uptake results in swelling, although the hydrated cellulose is insoluble even in hot water. Cellulose was used as an additive in a mixture with other polyols to obtain foams [13,14]; in specific cases 9–36% of lignin in composition with petroleum oil-derived polyol was applied to obtain rigid polyurethane foams [15]. Liquefying of cellulose in presence of glycerol also led to polyol, which was mixed with petroleum oil-derived polyols up to 70 mass% [16]. In other words, the addition of powdered cellulose fibers or cellulose derivatives like acetate, sulfate, or trimethylsilyl at the level of 11–44% into foaming composition enabled obtaining rigid PUFs with good mechanical properties and improved thermal resistance [17,18,19,20,21,22,23]. In the paper [24], the authors showed that cellulose can also be used for the synthesis of polyol, provided that it is initially swollen in water and GL is used for its hydroxyalkylation. The reaction of cellulose with glycidol can be conducted in 180 °C within 30 h in the presence of potassium carbonate as catalyst (Figure 3):

The polyol composed of hydroxyalkylated cellulose units has the viscosity 5538 mPa·s and density 1.283 g/cm^3^ at 20 °C. The PUF obtained from such polyol has higher thermal resistance than commercially used rigid PUFs. The PUF degrades in natural conditions up to 71 mass% within one month, while the polyol itself degrades completely [24].

Currently, in our studies we aimed at synthesis of the polyol from ecologically sustainable sources like metasilicic acid, cellulose, and ethylene carbonate to use it for obtaining PUFs with hypothetically improved thermal resistance, good mechanical properties, and biodegradability.

The here proposed polyol synthesis is simpler in comparison with other protocols using plant oils [25,26]. Commonly the oils need epoxidation of double bonds with organic peracids to increase the functionality of starting oils. That low yield conversion is followed by epoxide ring opening using diols. The polyols obtained in that way can be further used to obtain elastic PUF. The method elaborated here is a one pot reaction, without the necessity to isolate semiproduct. Thus, the obtained polyol is useful for obtaining rigid PUF of good thermal resistance and improved biodegradation in comparison to other rigid PUFs currently used in industry, which are very resistant to biodegradation. Thus, the products described here might be promising materials for practical use and biomass valorization.

## 2. Materials and Methods

### 2.1. Materials

The following materials were used in the work: cellulose (CEL, powder, 20 mesh particle size, 99% purity, Sigma-Aldrich, Taufkirchen, Germany), 40% water glass (DRAGON, Skawina Poland), hydrochloric acid (pure, POCH, Gliwice, Poland), GL (pure, Sigma-Aldrich, Taufkirchen, Germany), ethylene carbonate (EC, pure, Fluka, Buchs, Switzerland), potassium carbonate (pure, POCH, Gliwice, Poland), polymeric diphenylmethane 4,4′–diisocyanate (pMDI, Merck, Darmstadt, Germany), triethylamine (TEA, pure, Fluka, Buchs, Switzerland), surfactant Silicon L-6900 (pure, Momentive, Wilton, US), ethylene glycol, diethylene glycol, and triethylene glycol (pure Aldrich, Gillingham UK), and cyclohexanone (analytical grade S.A. POCH, Gliwice, Poland). Metasilicic acid (MSA) was obtained according to [1].

### 2.2. Synthesis of Metasilicic Acid

For this, 40% water glass was treated with concentrated hydrochloric acid to give white precipitate, which was filtered off and washed with a copious amount of water till neutral filtrate was obtained. The precipitate was dried at ambient temperature, and then at 80 °C to constant mass.

### 2.3. Synthesis of Polyols

The desired amounts of MSA, CEL, and GL (Table 1) were placed in a three-necked round bottom flask equipped with reflux condenser, mechanical stirrer, and thermometer. If necessary, water was added and the mixture was gradually heated to reach 120–140 °C temperature. The mixtures started to react with exothermic effect resulting in temperature increase as high as 240 °C. Then the mixture was heated at 180 °C for about 20–22 h, while CEL was observed to dissolve. After that, the mixture was cooled down to 80 °C and EC and potassium carbonate were added and the mixture heated at 145 °C or 160 °C–180 °C until the reaction was completed (Table 1). The end point of reaction was estimated based on mass balance; the mass of reaction mixture decreased due to decomposition of EC. The amount of unreacted EC was also determined analytically. The epoxide number (EN) and acidic number (AN) were also determined to analyze the amount of unreacted GL and MSA in the course of reaction.

### 2.4. Analytical Methods

The reaction of mixture MSA-CEL with GL was monitored by epoxide number determination using hydrochloric acid in dioxane [27]. The progress of reaction of hydroxyalkylation with EC was monitored using barium hydroxide method described in [28]. The sample was treated with 2.5 cm^3^ of 0.15 M barium hydroxide, vigorously shaken and the excess of barium hydroxide titrated off with 0.1 M HCl_aq_. The AN of polyol was determined by titration with 0.1 M sodium hydroxide in presence of phenolphthalein. Finally, the hydroxyl number (HN) of polyol was determined by acylation with acetate anhydride in dimethylformamide. Excess of anhydride was titrated with 1.5 M NaOH_aq_ in presence of phenolphthalein [29]. The ^1^H-NMR spectra of reagents were recorded at 500 MHz Bruker UltraShield instrument in DMSO-d6 and D2O with hexamethyldisiloxane as internal standard. IR spectra were registered on ALPHA FT-IR BRUKER spectrometer in KBr pellets or by ATR technique. The samples were scanned 25 times, in the range from 4000 to 450 cm^−1^ at 2 cm^−1^ resolution. MALDI ToF (Matrix-Associated Laser Desorption Ionization Time of Flight) spectra of oligomers were obtained on Voyager-Elite Perceptive Biosystems (US) mass spectrometer working at linear mode with delayed ion extraction, equipped with nitrogen laser working at 352 nm. The method of laser desorption from gold nanoparticles (AuNPET LDI MS) was applied [30]. The observed peaks corresponded to the molecular ions K+ (from catalyst) ions. The samples were diluted with methanol to 0.5 mg/cm^3^.

In order to identify side products in polyol, gas chromatography was used with cyclohexanone as internal standard. The gas chromatograph HP 4890A was used, equipped with HP-FFAP column of 30 m length, 0.25 mm diameter, 0.25 μm film thickness, and 260 °C port temperature and temperature profile: 40–220 °C, with 20 deg/min heating rate, the helium flow 18.3 cm^3^/min, and 0.2 μdm^3^ sample volume. A series of reference substances were used, i.e., ethylene glycol, diethylene glycol, triethylene glycol. The percentage of glycols in products was determined based on calibration curves with the same internal standard using Equation (1):(1)$$\frac{S_{g}}{S_{t}}=a\times \left(\frac{m_{g}}{m_{t}}\right)+b$$
where: *m_g_*_,_
*m_t_*—glycol mass or consecutive product of reaction with EC and mass of standard, respectively; *S_g_*, *S_t_*—integrated peak area of glycol or consecutive product and standard, respectively; *a*, *b*—experimental coefficients of calibration curves.

The mass of products obtained from EC and water and mass of products of consecutive reactions between glycol and EC (*m_g_*) were calculated from Equation (1). The percentage of side products (*p*) were calculated considering total sample mass (*m_p_*) according to Equation (2):(2)$$p=\frac{m_{g}}{m_{p}}100\%$$

Table 2 contains the determined coefficients of calibration curves of the analyzed glycols and their retention times.

### 2.5. Physical Properties of Polyol

The density, viscosity, and surface tension of the polyol were determined with pycnometer, Höppler viscometer (typ BHZ, prod. Prüfgeratewerk, Germany), and by the detaching ring method, respectively.

### 2.6. Polyurethane Foams

Foaming of polyol was performed in 500 cm^3^ cups at room temperature. The foams were prepared from 10 g of polyol, to which 0.4–0.5 g of surfactant (Silicon L-6900), 0.03–0.20 g of TEA as catalyst, and water (2–3%) as blowing agent were added. After homogenization, the polymeric diphenylmethane 4,4′-diisocyanate was added (8–21 g). The commercial isocyanate containing 30% of tri-functional isocyanates was used. The mixture was vigorously stirred until creaming began. The materials were then seasoned at room temperature for 3 days. The samples for further studies were cut from the obtained foam.

### 2.7. Properties of Foams

The apparent density [31], water uptake [32], dimensional stability in 150 °C temperature [33], thermal conductivity coefficient (IZOMET 2104, Bratislava, Slovakia), and compressive strength [34] of PUF were measured. Apparent density of PUFs was calculated as the ratio of PUF mass to the measured volume of PUF sample in a cube of 50 mm edge length. Water volume uptake was measured on cubic samples of 30 mm edge lengths. Dimensional stability was tested on samples of 100 × 100 × 25 mm size. Thermal conductivity coefficient was measured at 20 °C after 48 h of PUF seasoning. The needle was inserted 8 cm deep into a cylindrical PUF sample 8 cm in diameter and 9 cm high. Compressive strength was determined using burden causing 10% compression of PUF height related to initial height (in accordance with the PUF growing direction). Thermal resistance of modified foams was determined both by static and dynamic methods. In the static method, the foams were heated at 150, 175 and 200 °C with continuous measurement of mass loss and determination of mechanical properties before and after heat exposure. The 100 × 100 × 100 mm cubic samples were used to determine static thermal resistance and compressive strength. In the dynamic method, thermal analyses of foams were performed in ceramic crucible at 20–600 °C temperature range, about 100 mg sample, under air atmosphere with Thermobalance TGA/DSC 1 derivatograph, Mettler, with 10 °C/min heating rate. Topological pictures of PUFs were recorded for cross-sections of PUFs samples cut in direction perpendicular to growing. The pictures were analyzed with MORPHOLOGI G3 (Malvern Panalytical Ltd., Malvern, UK) using Morphologi software ver. 8.30 (Malvern Panalytical Ltd., Malvern, UK)

### 2.8. Flammability of Foams

Flammability of foams was determined by oxygen index [35] and horizontal test according to norm [36] as follows: the foam samples (150 × 50 × 13 mm) were weighed, located on horizontal support (wire net of 200 × 80 mm dimensions), and the line was marked at the distance of 25 mm from edge. The sample was set on fire from the opposite edge using Bunsen burner with the blue flame of 38 mm height for 60 s. Then the burner was removed and the time of free burning of foam reaching marked line or cessation of flame was measured by stopwatch. After that the samples were weighed again. The rate of burning was calculated according to the expression: (3)$$v=\frac{125}{t_{b}}$$
if the sample was burned totally, or using equation:(4)$$v=\frac{L_{e}}{t_{e}}$$
if the sample ceased burning, where:

*L_e_*—the length of burned fragment, measured as the difference 150 minus the length of unburned fragment (in mm). According to norms, if the burned fragment has 125 mm length, the foam is considered as flammable.

*t_b,_ t_e_*—the time of propagation of flame measured at the distance between starting mark up to the end mark or at the time of flame cessation.

The mass loss Δ *m* after burning was calculated from the formula:(5)$$\Delta \;m=\frac{m_{o}-m}{m_{o}}\cdot 100\%$$
where *m_o_* and *m* mean the sample mass before and after burning, respectively.

Flammability of foams was evaluated for samples 100 × 100 × 10 mm in size using a cone microcalorimeter, a product of Fire Testing Technology Ltd. (East Grinstead, UK), according to standard [37], by applying the heat flow 25 kW/m^2^ and the distance from ignition source 25 mm. During the tests, the time to ignition (TTI), total time of flaming (TTF), percentage mass loss (PML), heat release rate (HRR), effective heat of combustion (EHC), and total heat release (THR) were recorded.

### 2.9. Biodegradation of Polyol and Foam

The biodegradation of polyol and the PUF obtained from it was tested by the use of OxiTop Control S6 instrument (WTW-Xylem, Rye Brook, NY, USA). The respirometric method was used to measure the oxygen demand necessary for aerobic biodegradation of polymeric materials in soil. The measurement of consumed oxygen was presented using the value of biochemical oxygen demand (BOD), which is the number of milligrams of captured oxygen per mass unit of tested polyurethane material. The instrument was composed of six 510 cm^3^ glass bottles, equipped with rubber quivers and measuring heads, which were used to measure BOD. They allowed measurement of the pressure in the range of 500 to 1350 hPa with an accuracy of 1% at a temperature of 5 to 50 °C. The communication between the measuring heads and user was performed with Achat OC computer software (WTW-Xylem, Rye Brook, NY, USA), which was applied to interpret the obtained measurement results.

The biodegradation tests were performed according to the norm [38]. For a biodegradation test, sieved and dried gardening soil was used with the following parameters: 5% humidity (according to ISO 11274-2019 [39]), pH = 6 (according to ISO 10390-2005 [40]), and particle diameters <2 nm. The measurement was carried out in a system consisting of 200 mg of tested sample (oligomer or foam), 200 g of soil, and 100 g of distilled water. The samples were homogenized in bottles, rubber quivers containing 2 pastilles of solid NaOH were mounted and sealed with measuring heads for six samples. The set was incubated at 20 ± 0.2 °C for 28 days. The current oxygen consumption was determined within 2–3 days intervals for samples and two references: positive and negative, plus blank which was the soil and water only. The starch was used as positive sample, while polyethylene was the negative sample. *BOD* was determined for every sample taking into account the *BOD* of the tested system reduced by the *BOD* of the soil and concentration of the tested compound in the soil using the following formula:(6)$$BODs=\frac{BOD_{x}-BOD_{g}}{c}$$
where: *S*—number of measurement (in days); *BOD_S_*—biochemical oxygen demand of the analyzed sample within S days (mg/dm^3^); *BODx*—biochemical oxygen demand of the measuring system (bottle with sample and soil) (mg/dm^3^); *BODg*—biochemical oxygen demand of soil without a sample (mg/dm^3^); and *c*—sample concentration in the tested system (mg/ dm^3^).The degree of biodegradation of the oligomer mixture or foam based on it was determined using the formula:(7)$$D_{t}=\frac{BOD_{S}}{TOD}\cdot 100\%$$
where: *Dt*—biodegradation degree of sample (%); *TOD*—theoretical oxygen demand (mg/dm^3^).

The theoretical oxygen demand was calculated using the formula given in norm ISO17556-2019 [39]. It has been assumed that in oxygen conditions, the carbon is converted into CO_2_, hydrogen into H_2_O, and nitrogen into NH_3_.

For the compound of known *C*, *H*, *N*, *O*, and *Si* percentage and total mass of sample, the *TOD* value can be calculated from the following equation:(8)$$TOD=\frac{16\cdot \left[2C+0.5\cdot \left(H-3N\right)+2Si-O\right]}{m}$$
where: *C*, *H*, *N*, *O*—mass fractions of elements in biodegraded material; *m*—the sample mass of material (g).

## 3. Results and Discussion

### 3.1. Preparation of Polyols

Recently we have elaborated the routes to liquefy the MSA [1] and CEL [24] by hydroxyalkylation and obtained the polyols from these substrates. We have now attempted to use both substrates in one step to obtain a polyol. The starting point was the optimization of reagents for such synthesis, i.e., MSA, CEL, GL, and EC, according to the following criteria: (i) the polyol has to be miscible with liquid pMDI used to prepare the foaming composition; (ii) the polyol should be suitable to obtain rigid PUF, which is expected for short oxyalkylene chains of polyol, while the percentage of MSA and CEL should be kept high; (iii) targeting PUF with high compressive strength and thermal resistance, which is expected with a high percentage of MSA in PUF; and (iv) considerable biodegradability of PUF, which is achievable with high CEL percentage in PUF. Preliminary syntheses of polyols (Table 1, No. 1 and 2) were performed using equal amounts of CEL and MSA, or using twice MSA related to CEL, the latter serving to ensure good mechanical properties of the PUFs obtained therefrom. We have determined the viscosity of the polyols to estimate their miscibility with liquid pMDI. However, the obtained polyols were semi-solid resins, not miscible with pMDI. Varying the amount of EC (Table 1, No. 3–4) did not improve considerably the liquefying of eventual polyol. Moreover, the reactions to obtain polyol were accompanied by the unfavorable high exothermic effect resulting from the reaction of GL with MSA. When 5% water was introduced into the reaction mixture, the exothermic effect of reaction was quenched by water (Table 1, No. 5–8). In such a way, two polyols (6 and 7) were obtained which were miscible with pMDI. They were then used as polyol component in foaming process with pMDI. The PUFs obtained therefrom were then tested for their thermal resistance by heating for 24 h at 150 °C (Table 3). Apparent densities of obtained PUFs were within that determined for standard rigid PUFs. The best results were obtained for the PUF obtained from polyol 7 (Table 3) and we have scaled up the synthesis of this polyol (Table 1, No. 8–10). In synthetic procedure we have shortened the reaction time with EC from 18 h at 145 °C to 6 h at 180 °C. After this step, we determined AN of polyol and found it equalled zero, indicating that MSA was consumed totally, and we determined the amount of EC, which was 0.06% and revealed total consumption of EC in this process.

### 3.2. Composition and Structure of Polyols

The following reactions took place in the synthetic process: hydroxyalkylation of MSA (Figure 1), CEL (Figure 3), and water upon addition of GL, and in the second step the reaction of obtained semi-products with another hydroxyalkylating agent, EC, as illustrated in Figure 4.

Thus, the final product is actually the mixture of three products of hydroxyalkylation of: CEL, MSA, and water. The reaction was monitored by IR and ^1^H NMR spectroscopies by comparison with spectra of substrates. The products were also characterized in detail by chromatography and MALDI-ToF analysis.

The IR spectrum of MSA is presented at Figure 1a. The broad band in the 3500–3000 cm^−1^ region is characteristic for O-H stretching vibrations of hydrogen bonding engaged hydroxyl groups. Moreover, that the valence Si–OH band overlapped Si–O–Si at 1100 cm^−1^ was observed. In the IR spectrum of CEL (Figure 1b), the hydroxyl group bands occur at 3350 cm^−1^, while primary and secondary hydroxyl C-OH bands are observed at 1058 cm^−1^ and 1165 cm^−1^, respectively. Hydroxyl groups deformation bands were found in the 1280–1340 cm^−1^ region and at 614 cm^−1^. The band at 2900 cm^−1^ was attributed to methylene and methine groups, and also the deformation band of methylene groups was found at 1430 cm^−1^. In the IR spectrum of the polyol (Figure 1c), the broad band in the region 3500–3000 cm^−1^, which was present in the spectrum of CEL, decreased; instead, a lower intensity band from the hydroxyl groups of the product of hydroxyalkylation of MSA and CEL replaced it. In the spectrum of the polyol, the valence Si–OH band at 1100 cm^−1^ disappeared, replaced by the C-O-C ether band from opened rings of GL and EC, overlapped with C-O-Si band. On the shoulder of the latter, the C-OH band at 1100 cm^−1^ is present. The presence of C-O-Si fragments was confirmed by the presence of the band centered at 934 cm^−1^. Methylene and methine deformation bands are visible at 1454 and 1328 cm^−1^, respectively. No carbonyl band at 1800 cm^−1^ was observed in the spectrum of the polyol, evidencing the absence of unreacted EC. 

In the ^1^H-NMR spectrum of the polyol (Figure 2), no resonances at 3.6 and 3.0 ppm characteristic for epoxide rings were observed, which evidenced that the polyol is free of unreacted GL. 

Neither were the resonances of free EC at 4.5 present in the spectrum of polyol, indicating unreacted EC was absent in the polyol [41]. The resonances in the 3.2–3.7 ppm region were attributed to methylene and methine groups which emerged due to ring opening of hydroxyalkylating agents, namely, GL and EC as well, and contained in cellulose units. The coalesced one hydroxyl group proton resonance was present at 4.6, which disappeared upon selective deuteration with D_2_O.

The composition of the polyol could be determined in detail by MALDI-ToF spectrometry (Table 4, Figure 3). The trace amounts of substrates were identified with low *m*/*z* (Table 4, entries 1, 2). Monomeric and oligomeric MSA fragments of the general formula (H_2_SiO_3_)_n_ were found *(n* = 2 or 3; Table 4, entries 9, 15, 24, 28, 32). The analysis of MALDI-ToF evidenced that also the trace products of hydroxyalkylation of MSA with GL (Table 4, entries 5, 19, 25, 30) and EC (Table 4, entries 12, 17, 18, 29, 31) were present. The products of the addition of GL to MSA and further addition of EC were identified as adducts with K^+^ or as the dehydrated species (Table 4, entries 11, 18, 24, 29, 31). The series of peaks differing by *m*/*z* = 44, corresponding to oxyethylene fragment, illustrate the stepwise hydroxyalkylation (Table 4, entries 12, 17). 

Moreover, the products of reaction between water and GL (Table 4, entries 7, 14, 26, 33) and EC (Table 4, entry 11) were found as well as the products of GL oligomerization (Table 4, entries 8, 22, 34). The products of oxyalkylation of GL with EC and products of reaction between oligomers GL with EC were identified (Table 4, entries 4, 13, 16). High temperature of synthesis with EC probably is responsible for dehydration of hydroxyalkyl chains and presence of side products (Table 4, entries 18, 19, 29) according to Figure 5.

Reaction between water and EC can lead to formation of ethylene glycol, diethylene glycol, and triethylene glycol according to Figure 6.

The semi-quantitative analysis of low molecular weight products was performed by chromatographic method (Table 2). It has been found that small amounts of glycols (1.00% ethylene glycol, 0.70% diethylene glycol, and 0.24% triethylene glycol) were present in the polyol obtained from MSA and CEL, which in total render this product as containing 2% glycols, in contrast to previously obtained polyols from MSA only, which contained almost 20% glycols [1]. 

The physical properties, like density, viscosity, and surface tension of the polyol were determined to be: 1.307 g/cm^3^, 26,560 mPa·s, and 51 mN/m. Typical PUF-suitable polyols have a viscosity within 200–30,000 mPas [42]. From our previous experience, we have learned that suitable surface tension of PUF-suitable polyols should be within 30–50 mN/s. Low surface tension is preferred because it enables effective foaming process. The obtained polyols have relatively high surface tension, which imposes the need for a relatively larger amount of surfactant in foaming composition. The HN (765 mg KOH/g) of the polyol suggested its suitability to obtain rigid PUFs.

### 3.3. Preparation and Properties of Polyurethane Foams

The PUFs were obtained in reaction of polyol with pMDI and water in presence of TEA as catalyst and SiliconL-6900 as surfactant. A couple of foaming attempts were executed in order to optimize the properties of PUF according to the criteria of a rigid PUF and small uniform pores, and the results are collected in Table 5. The amount of catalyst, isocyanate, and foaming agent were optimized. The foaming agent was carbon dioxide, which was evolved upon reaction of water with pMDI. The cream, rise, and tack-free times were observed during the foaming process.

The water was optimized at the level of 2–3%; at lower water percentages, the PUFs had high density, while at >3% water the formed PUFs were fragile. The amount of pMDI was crucial. The optimum amount of pMDI was found with isocyanate coefficient (IC) between 1.0 and 1.1. When the foaming composition with >1.0 IC was fixed and 2% water was used, the obtained foams were under-crosslinked, with sticky surface. High HN of polyol imposed that a low amount of catalyst was needed to initiate the crosslinking and growth of PUF. The optimized catalyst percentage was 0.5 or 0.3% related to mass of polyol if water amount was kept 2 or 3%, respectively. With a higher percentage of catalyst, the PUFs broke and pores were stretched. The cream and rise times of optimized compositions 6 and 11 depend on the amount of water in foaming composition. In the case of 2% water, they were equal, 60 and 50 s, respectively. The PUFs dried immediately. When 3% water was used, the cream time was elongated twofold, the rise time was also slightly longer and tack-free time grew considerably.

The following physical properties of PUFs were determined: apparent density, water absorption by soaking, dimensional stability, heat conductance coefficient, pore size, thermal resistance, compressive strength, and glass transition temperature. The apparent density of PUFs was dependent on the amount of water in foaming composition. At the higher limit (3% water), more CO_2_ was evolved resulting in better foaming and eventually low density PUFs were formed, with 51 kg/m^3^ value. At the lower water limit (2%), the apparent density of PUF was 72 kg/m^3^ (Table 6). Water uptake of PUFs did not exceed 2.8% after 24 h soaking PUF in water. This suggested that closed pores are present in the PUF, which is a desired structural property for potential use as a thermal insulator. This was later confirmed by heat conductance coefficient (0.0242–0.0244 W/mK), falling into the region characteristic for rigid PUFs [42]. Obtained PUFs have good dimensional stability, not exceeding 0.8%.

The physical properties of PUFs obtained here were compared with the properties of PUFs from MSA-derived polyol (MSA-GL-EC) and CEL-derived polyol (CEL-GL-EC) described previously [1,24]. The polyol obtained from mixture of MSA and CEL resulted in a decrease of apparent density of the obtained PUF in comparison with that obtained from MSA (Table 6), and a slight increase of density related to that of the CEL-based PUF (comp. 6). 

The water uptake of the PUF obtained from MSA/CEL mixture decreased by 2–3 times, while the heat conductance decreased as well if compared with those for PUFs obtained from MSA and CEL alone, the latter reaching the values requires for heat insulating materials.

Pore size of obtained PUF depended on water percentage in foaming mixture. The pore size increased along with water percentage. This was due to larger amount of CO_2_ released upon reaction of water with isocyanate groups of pMDI. The pores were oval shaped with 200–250 µm and 300–330 µm diameter for foaming mixtures containing 2 and 3% water, respectively. Maximal standard deviation of pore size was below 26%. As has been mentioned, the low water uptake (2.8%) indicates the presence of closed pores in PUF. The properties of PUFs changed upon heating. After one month heating at 150 °C, the pores were partially broken (Figure 4) due to increasing CO_2_ inside the pores. Annealing PUF at 175 °C resulted in further breaking pores, while annealing PUF at 200 °C resulted in graphitization of PUF.

One month heating showed rather low mass loss during annealing at 150 °C and greater upon heating at 175 °C (7.2–7.7% and 22.3–23.9%, respectively, Table 7). For comparison, the classic rigid PUFs show one day mass loss equal to 21 and 37%, respectively [43]. A considerable increase of thermal resistance for the obtained PUFs if compared with that of solely MSA- and CEL-based PUFs was observed (Table 7). Furthermore, the PUF obtained from MSA-based polyol is not resistant to thermal exposure at 200 °C, in contrast with the obtained MSA/CEL-based PUFs. 

The macroscopic pictures of PUFs before and after thermal exposure are shown in Figure 5. The color of PUF changed from light-yellow to brown along with temperature increase. The deformation of PUF was observed at 200 °C temperature.

Annealing of PUF resulted in mass loss of samples, mostly in first day of heating (Figure 6).

The mass loss of obtained PUFs is comparable to that of PUFs containing azacyclic units [44], which were incorporated deliberately to enhance thermal resistance of PUFs. The latter showed mass loss equal to 6–7% and 20% in comparable conditions. At 200 °C, the mass loss of obtained PUFs was within 32–39%, which is still comparable to those containing azacyclic units, which is no higher than 40%.

The compressive strength of obtained PUFs depends on water percentage in foaming compositions. The PUF obtained with 2% water in composition has typical compressive strength for rigid PUF, while that obtained with 3% water in composition is less mechanically resistant. Generally, the obtained PUFs showed increased compressive strength upon thermal exposure at 150 °C and 175 °C than before annealing (Table 7). This is due to additional crosslinking during thermal exposure. Eventually, the compressive strength of PUF obtained with 2% water in composition increased 1.5 times after annealing at 150 °C and fourfold upon heating at 175 °C in comparison with samples not treated thermally. Thermal exposure at 200 °C resulted in a slight decrease of compressive strength due to degradation of polymer.

Thermal analysis by dynamic method indicated that the PUF obtained with 3% water in composition started to lose mass at 236 °C, while that obtained with 2% water started to decompose at 139 °C (Table 8); the latter can be related to higher absorption of water from air by the foam. This suggestion agreed well with the fact that both PUFs show the same temperature for 10% mass loss; at this temperature the water absorption does not influence the processes accompanying mass loss. Such an hypothesis was then confirmed by water uptake studies (Table 6). The TG and DTG profiles showed that fast decomposition occurred at 280 °C and 320 °C (Figure 7). The first event corresponds to thermal dissociation of urethane bonds, while decomposition of polyurethane into amines and CO_2_ corresponds to the second peak [45]. The third exothermic peak at 390 °C is related to thermal decomposition of CEL units. Total decomposition of PUF ended at temperature ca 600 °C. The residua mass after thermal decomposition was 14–17% of initial mass. Glass transition temperatures of obtained PUFs were 68 and 88 °C (Table 8), which categorized the PUFs as rigid ones.

The obtained PUFs were tested for flame resistance (Table 9). Horizontal flaming tests indicated that the PUFs are flammable; the flaming reached the full length of sample (150 mm) after ignition, the flaming rate being dependent on apparent density. The PUF of lower density flamed faster (1.3 mm/s) than the cast PUF (of higher density) (1.0 mm/s). Obtained PUFs have diminished flammability compared with those obtained from MSA-based polyol, which showed 2 mm/s flame rate, and those obtained from CEL-based polyol, which flamed totally.

The annealed PUFs have diminished flammability, which is an advantageous feature for foams that are designed to operate at high temperatures. A one-month thermal exposure at 150 °C converts the PUFs into self-extinguishing ones. The flame ceases at 20–25 mm distance and does not reach the end line. The PUFs annealed at 175 °C did not ignite upon contact with flame (Table 9). The reduced flammability was confirmed by oxygen index. Not annealed PUFs showed 19.9–20.2%, the PUFs annealed at 150 °C had 21.5%, while the PUFs annealed at 175 °C had above 29% oxygen index, which allows the classification of them as non-flammable materials [46]. This suggests that obtained materials can be used in elevated temperatures.

This reduced flammability is confirmed by the change in the elemental composition of foams, in which the content of hydrogen decreases during heating (the presence of which is one of the causes of burning organic compounds), and the content of nitrogen (an element that hinders combustion, Table 10) increases. The IR spectra of annealed PUFs showed some characteristic changes as well. In the IR spectrum of starting PUF (Figure 8), the broad band at 3300–3600 cm^−1^ from stretching N-H vibrations was identified, accompanied by bending NH at 1600 cm^−1^, and C-H stretching vibration bands were observed. The isocyanate group band at 2280 cm^−1^, carbodiimide band at 2136 cm^−1^, and carbonyl band at 1710 cm^−1^, were found in the spectrum of not-annealed PUF. After thermal exposure of PUF, the characteristic changes in IR spectra were observed, namely, disappearance of carbodiimide band, presumably as a result of its oxidation, and also disappearance of residual isocyanate band. The latter is probably related to additional crosslinking between residual isocyanate groups and hydroxyl groups from polyol. Some oxidation processes must have occurred which could be spectrally identified by the increase of intensity of the bands centered at 3400 cm^−1^ and 1300 cm^−1^ from O-H and C-OH, respectively. Furthermore, in the IR spectra of the annealed PUFs the intensity of the band centered at 1660 cm^−1^ suggested that C=C bonds are formed in thermally converted polymer. This process of graphitization might be responsible for achieving for the material a higher compressive strength and also diminished flammability. In the IR spectra of annealed PUFs, the valence C-H and deformation =C-H bands are observed at 3300−2500 cm^−1^ and 1000−750 cm^−1^, respectively. The C-O-Si remain untouched at 1100 cm^−1^ and 934 cm^−1^_._

Using cellulose as substrate to obtain polyol and further PUF holds promise for obtaining biodegradable materials. We have tested the obtained polyol and PUF obtained from it in soil under air access. The BOD profile is shown at Figure 9. The elemental composition of samples was available from elemental analysis (Table 11). Based on the elemental composition we have calculated theoretical biological demand (TOD). The level of degradation (Dt) of samples was determined within 28 days in reference soil environment (Table 12). We have found that PUF undergoes easy degradation up to 46 mass%. This is lower than previously reported biodegradation for PUF obtained solely from CEL-based polyol (71%); however, the PUF is a material that biodegrades well considering the environmental impact of the obtained PUF.

## 4. Summary and Conclusions

The synthesis of a polyol based on cellulose and metasilicic acid was elaborated in a one pot reaction.The polyol with incorporated silicon and oxyalkylated cellulose was obtained as substrate to obtain new polyurethane foams.Obtained polyurethane foams showed low water uptake, high dimensional stability at elevated temperatures, regular structure of pores, and low heat conductance coefficient, which renders the polymer a good candidate for use as a heat insulating material.Obtained polyurethane foams show enhanced thermal resistance. They can stand long term heating at 175 °C. The rigid foam has good mechanical properties; its compression strength grew after one-month thermal exposure at 150 and 175 °C by 60% and ca 300% of initial value, respectively.The polyurethane foams were obtained based on environmentally friendly substrates, namely, widespread plant material—cellulose, biologically neutral metasilicic acid, alkylene carbonates (which are considered as green chemicals), and non-toxic, non-volatile diphenylmethane diisocyanate.The polyurethane foams from polyols based on metasilicic acid and cellulose are biodegradable up to 46% within one month, while polyol substrates are 100% biodegradable according to standard soil test.

## Data Availability

Not applicable.

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
