# Peer review of "Polyols and Polyurethane Foams Obtained from Mixture of Metasilicic Acid and Cellulose"

_polymers, 2022, doi:10.3390/polym14194039_

Round 1

Reviewer 1 Report

Dear authors,

The article “Polyols and polyurethane foams obtained from a mixture of 2 metasilicic acid and cellulose” needs corrections. The article has no deep problems with the content, but the presentation of the results is deficient.

Introductions

Make it clear what progress the proposed one contributes compared to other polyols from renewable or biodegradable sources.

Materials

Define the acronyms in this section. Although the definitions have already been made in the text, it is essential to have them here in case any reader wants to repeat the work.

Synthesis section

The authors should clarify the data presented in Table 1. It is known that polyol and MDi are not miscible. So, do the authors mean by not miscible with pMDI? This should be made more explicit in the reaction comments. This explanation should be supplemented, though explained in lines 264 and 265.

In experimental procedures, more should be left as it was done. For example, what FTIRs were taken in what scan range? Resolution. See an example: https://doi.org/10.1002/app.50709

In section 2.6, you can leave only “Polyurethane foam”

Results

Regarding the FTIR results, the spectra must clarify the appropriate bands. They must be identified. See the example: https://doi.org/10.1002/app.50709

Figures 1 a), b) and c) can be grouped into just one figure. The authors should improve the resolution of the figure. Note that the x-axis is on different scales.

NMR results should be improved and presented differently. See the example: https://doi.org/10.1016/j.molliq.2019.04.078

Also, the x-axis decimal separator must be changed to a period instead of a comma.

How is the silicon-bonded? Is the condensation on silanol always the same?

Is there a way to identify differences in the hydroxyl groups? Si-OH and –CH2-OH?

In figure 3 the correlation coefficients must be included together with the adjusted models.

What was the criterion for choosing the adjustment functions in figure 3?

In figure 4, the caption must be corrected to. “Optical microscopy of foam...” In discussing these results, it is recommended to comment in the text on the sizes of the foam cells and their standard deviation, as well as if there is more formation of open or closed pores.

In figure 6, the Y axis decimal separator must be corrected, and the respective unit must be included.

In Figure 7, its presentation and inclusion of the respective bands in this Figure should be significantly improved. See an example: https://doi.org/10.1002/app.50709

The authors have an exciting work, but overall they should improve the presentation of figures and equations. The presentation of information is essential to win over the readers of the work. I recommend a review and more outstanding care in constructing the figures. It is attractive as a polyol option and biomass valorization, which should be included in the introduction.

Reviewer 2 Report

Dear Editor, Dear authors,

The article presented by Lubczak et al. is a very interesting study about alternative bio-based polyurethane which has to be considered for publication. The preparation of the material so as its analysis is presented very extensively considering many aspects from chemistry, to mechanical properties, to biodegradability. Therefore, this contribution has a high scientific impact but unfortunately it suffers of several drawbacks due to complexity of the reporting.

Here underneath the list of the major observations that I have detected:

-        Macroscopic picture of the more interesting foams must be presented.

-        Indicate the amount of pMDI which was used for foaming.

-        In order to quickly find the acronyms, please organize the paper so that all acronyms are explained in the material part (in intro you could report the full name).

-        Graphical elaboration of FT-IR and NMR is very poor. It looks like an old book. Please present the 3 FT-IR spectra in a single graphic. NMR quality is not acceptable. Analytical details for both techniques need to be extended in the experimental section.

-        Elaboration of fig.7 is better, but the three spectra need to be shifted so that the trend can be compared.

-        The interpretation of MALDI-ToF for this kind of polymers make sense if the authors can determine repeating pattern at different m/Z. For instance, if you have signals at 20, 45, 70, 95 and so on (ideally with similar surrounding), it could be reasonably expected that a repeating fragment of 25 m/Z belong to the polymer. For determining this, the MALDI spectra should be reported.

Indeed, the attribution of the single peaks in a table can easily be achieved by combination of other units.

-        Scheme 6 is missing the side products (CO2 I guess) and e.c.t. is unclear (maybe should be etc?)

-        Figure 3 is not recall in the text and very shortly considered. The author can remove this part or discuss this result more into details, maybe comparing this trend with other formulation (by the way… which formulation was this?).

-        Figure 4 are not very sharp (with optical microscope is hard to do better…) there is any chance to observe the foams at the SEM?

-        Chemical molecules should be presented with the same elaboration program (e.g. scheme 1, 4 and 6 are very different).

-        Some typos are present in the article: Table 6 : Height; Table 5: Cracks

In Table 5, line 415, line 510, line 556:  space missing.

This indication may serve the authors to further improve the homogeneity of their consistent work which I would be pleased to receive in the next revision round.

Round 2

Reviewer 1 Report

Dear authors

In figure 2 I expected the authors to include an image of the structure and start the protem they are referring to, as in the example suggested.

Figure 5 has been included, a scale must be included to know the size of the samples in the image.

They must check the format of the references

Reviewer 2 Report

Dear Editor, Dear authors, 

The quality of the presentation is increased, not perfect yet, but I understand the difficulties.

I recommend acceptance after polishing (that could be done during editing) and in particular:  The font of the text must respect the template of Polymers - especially in the reference section.
